# Gene Profiling of a 3D Psoriatic Skin Model Enriched in T Cells: Downregulation of *PTPRM* Promotes Keratinocyte Proliferation through Excessive ERK1/2 Signaling

**DOI:** 10.3390/cells11182904

**Published:** 2022-09-16

**Authors:** Geneviève Rioux, Florence Turgeon, Gaëtan Le-Bel, Camille Grenier, Sylvain L. Guérin, Roxane Pouliot

**Affiliations:** 1Centre de Recherche en Organogénèse Expérimentale de l’Université Laval/LOEX, Axe Médecine Régénératrice, Centre de Recherche du CHU de Québec-Université Laval, Québec, QC G1J 1Z4, Canada; 2Faculté de Pharmacie, Université Laval, Québec, QC G1V 0A6, Canada; 3Centre Universitaire d’Ophtalmologie-Recherche, Centre de Recherche du CHU de Québec-Université Laval, Axe Médecine Régénératrice, Québec, QC G1S 4L8, Canada; 4Département d’Ophtalmologie, Université Laval, Québec, QC G1V 0A6, Canada

**Keywords:** psoriasis, tissue-engineered skin, ERK1/2, PTPRM, transcriptome, keratinocytes, microarray

## Abstract

Psoriasis is a complex, immune-mediated skin disease involving a wide range of epithelial and immune cells. The underlying mechanisms that govern the epidermal defects and immunological dysfunction observed in this condition remain largely unknown. In recent years, the emergence of new, more sophisticated models has allowed the evolution of our knowledge of the pathogenesis of psoriasis. The development of psoriatic skin biomaterials that more closely mimic native psoriatic skin provides advanced preclinical models that will prove relevant in predicting clinical outcomes. In this study, we used a tissue-engineered, two-layered (dermis and epidermis) human skin substitute enriched in T cells as a biomaterial to study both the cellular and molecular mechanisms involved in psoriasis’ pathogenesis. Gene profiling on microarrays revealed significant changes in the profile of genes expressed by the psoriatic skin substitutes compared with the healthy ones. Two genes, namely, *PTPRM* and *NELL2*, whose products influence the ERK1/2 signaling pathway have been identified as being deregulated in psoriatic substitutes. Deregulation of these genes supports excessive activation of the ERK1/2 pathway in psoriatic skin substitutes. Most importantly, electrophoresis mobility shift assays provided evidence that the DNA-binding properties of two downstream nuclear targets of ERK1/2, both the NF-κB and Sp1 transcription factors, are increased under psoriatic conditions. Moreover, the results obtained with the inhibition of RSK, a downstream effector of ERK1/2, supported the therapeutic potential of inhibiting this signaling pathway for psoriasis treatment. In conclusion, this two-layered human psoriatic skin substitute enriched in T cells may prove particularly useful in deciphering the mechanistic details of psoriatic pathogenesis and provide a relevant biomaterial for the study of potential therapeutic targets.

## 1. Introduction

Psoriasis is a complex cutaneous disorder characterized by chronic inflammation accompanied by alterations in the proliferation and differentiation of epidermal cells, in addition to the involvement of immune cells that participate to the inflammatory tissue responses. The crosstalk between the cells involved generates an inflammatory microenvironment that supports the establishment of psoriatic features, such as leukocytes infiltration and an increased production of soluble active mediators [1]. Keratinocytes play an active role in the pathogenesis of psoriasis by taking part in several processes such as hyperproliferation, impaired differentiation, and feedforward-acting proinflammatory cytokines release [2,3]. However, the underlying mechanisms regulating these epidermal defects and the immunological dysfunction remain largely unknown. In recent years, the emergence of new and more sophisticated models has allowed the evolution of our knowledge of the pathogenesis of psoriasis [4,5,6]. The development of psoriatic skin biomaterials that more closely mimic the native psoriatic skin provides advanced preclinical models that will prove relevant in predicting clinical outcomes, thereby decreasing inaccurate predictions that can lead to adverse events.

In the past, our team investigated the transcriptomic profile of our psoriatic skin biomaterial produced from psoriatic patient skin cells [7,8]. This 3D model only included pathological fibroblasts and keratinocytes but still provided a 54% match in the pattern of expressed genes with other profiling studies conducted on native psoriatic skin, despite the absence of the multiple cell types that are normally present in the native skin [7]. Additionally, according to Niehues and colleagues, this tissue-engineered psoriatic model is the one displaying the greatest number of psoriatic features, including acanthosis, parakeratosis, hyperproliferation and impaired keratinocyte differentiation [9]. These results demonstrate the reliability of our biomaterial and put forward the contribution of these two cell types in psoriasis, i.e., fibroblasts and keratinocytes. In the last decade, our team has also been focusing on adding the immune component to the tissue-engineered psoriatic skin model [4,10,11,12,13]. Recently, we successfully developed a T cell-enriched skin model produced from the skin cells of psoriatic patients, which exhibited an inflammatory microenvironment representative of the pathology, including increased production of IL-17A, IFNƴ, CCL2, CXCL10, IL-6, CXCL8 and IL-1ra compared with the healthy skin model [4]. This new model represents a most interesting tool to further investigate the mechanisms involved in psoriasis.

Signal transduction through intracellular protein kinases allows keratinocytes to maintain a balance between numerous cellular functions, including differentiation, cytokine expression, apoptosis, and proliferation. Mitogen-activated protein kinases (MAPKs) are among the most extensively characterized intracellular signaling pathways components. MAPKs relay, amplify and integrate signals from a wide range of stimuli and induce appropriate physiological responses through three major metabolic pathways: extracellular signal-regulated kinases (ERK1/2), c-Jun N-terminal kinases (JNK) and p38 [14,15]. Upon activation by the phosphorylation of both threonine and tyrosine residues by upstream regulators, each of these MAPKs regulates downstream targets, including intracellular enzymes and transcription factors [16]. Excessive MAPK activation is a common cause of many diseases. For instance, persistent activation of the p38 and JNK signaling pathways has been suspected to mediate neuronal apoptosis in various diseases, including Alzheimer’s and Parkinson’s, while the ERK1/2 signaling pathway appears to be central to multiple stages of tumorigenesis, especially proliferation, migration and invasion of cancer cells [17,18,19]. Furthermore, psoriatic keratinocytes are reported to have abnormalities in the activation of various MAPK signaling pathways [20,21]. The contribution of the ERK1/2 signaling pathway to cell proliferation and survival makes it a target of interest for better understanding the pathogenesis of psoriasis [22].

In the current study, we highlighted the most deregulated genes between healthy and psoriatic T cell-enriched, tissue-engineered models. Two gene-encoding proteins that influence the ERK1/2 signaling pathway, namely, *PTPRM* and *NELL2*, could then be identified. The deregulation of these genes supported the excessive activation of the ERK1/2 pathway typically observed in psoriatic skin substitutes. Electrophoresis mobility shift assays (EMSAs) provided evidence that the DNA-binding properties of two ERK1/2 downstream nuclear targets, namely, the transcription factors nuclear factor-kappa B (NF-κB) and specificity protein 1 (Sp1), are increased under psoriatic conditions. Moreover, the results obtained with the inhibition of RSK, a downstream effector of ERK1/2, supported the therapeutic potential of inhibiting this signaling pathway for psoriasis treatment.

## 2. Materials and Methods

This study was conducted in agreement with the Helsinki declaration and was performed under the guidelines of the Research Ethics Committee of the CHU de Québec-Université Laval (ethic code: DR-002-1121 and DR-002-1387, both renewals approved on 19 January 2022). All donors were given adequate information to provide written consent.

### 2.1. Patients and Biopsies

The subjects were all Caucasian females and males. Healthy human fibroblasts and keratinocytes were isolated from breast reduction skin biopsies from 3 healthy donors (18-, 46- and 49-years old). Psoriatic human fibroblasts and keratinocytes were isolated from a 6 mm biopsy punch taken directly from the lesions of 4 psoriatic patients (36-, 46-, 49- and 64-years old) with chronic plaques of psoriasis vulgaris. Fibroblasts and keratinocytes were extracted from these biopsies according to a protocol described elsewhere, which is based on digestion using thermolysin followed by digestion using trypsin or collagenase [23]. Psoriatic patients received the same treatment prior to biopsy (methotrexate), and all patients had between 10 and 20% of their body surface area affected.

### 2.2. Cell Culture and Media

All cells were cultured in an incubator at 37 °C under 8% CO_2_. Fibroblasts were cultured in the Dulbecco–Vogt modification of Eagle’s medium (DMEM) supplemented with 10% bovine growth serum (HyClone, Thermo Fisher Scientific, Ottawa, ON, Canada), 60 µg/mL penicillin G (Sigma-Aldrich, Oakville, ON, Canada) and 25 µL/mL gentamicin (Schering, Pointe-Claire, QC, Canada). Keratinocytes were cultured in a combination of DMEM with Ham’s F12 (3:1), supplemented with 5% Fetal Clone II serum (HyClone, Thermo Fisher Scientific), 5 µg/mL insulin (Sigma-Aldrich), 0.4 µg/mL hydrocortisone (Galenova, St-Hyacinthe, QC, Canada), 10^−10^ M cholera toxin (MP Biomedicals, Montreal, QC, Canada), 10 ng/mL human epidermal growth factor (EGF; Austral Biological, San Ramon, CA, USA), 60 µg/mL penicillin (Sigma-Aldrich) and 25 µg/mL gentamicin (Schering).

### 2.3. T cell Culture and Polarization

T cell culture was performed using modified protocols [4,24,25,26]. Briefly, T cells were isolated directly from anticoagulated peripheral blood obtained from healthy donors by negative immunomagnetic selection. For this step, the EasySep™ Direct Human T Cell Isolation Kit (StemCell Technologies, Vancouver, BC, Canada) was used according to the manufacturer’s instructions, under sterile conditions and at room temperature. T cells were then cultured for 7 days at 37 °C under 8% CO_2_ in the presence of various cytokines to induce their polarization into Th1 and Th17 cells. During the first 3 days, T cells were seeded in 12-well plates at 2 × 10^6^/mL with their respective cytokine mixtures. Th1 polarization was achieved by adding IL-12 (20 ng/mL; StemCell Technologies, Vancouver, BC, Canada) and anti-IL-4 (5 µg/mL; Invitrogen, Waltham, MA, USA) while anti-IFNγ (10 µg/mL; Invitrogen, Waltham, MA, USA), anti-IL-4 (5 µg/mL), IL-1β (20 ng/mL; StemCell Technologies) and IL-6 (40 ng/mL; StemCell Technologies) were added to the medium for Th17 polarization. After these 3 days, Th1 and Th17 were separately exposed to 25 ng/mL of phorbol 12-myristate 13-acetate (PMA; Sigma, St-Louis, MO, USA) and to 1 µg/mL of ionomycin (Sigma) for 4 h at 37 °C. Following this stimulation, T cells were reseeded (2 × 10^6^/mL) in 12-well plates for an additional 4 days at 37 °C under 8% CO_2_ with IL-2 (30 U/mL R&D Systems, Burlington, ON, Canada) for Th1 cells and with IL-2 (10 U/mL) and IL-23 (20 ng/mL; Biolegend, San Diego, CA, USA) for Th17 cells.

### 2.4. Production of Tissue-Engineered Skin Substitutes

Healthy and psoriatic skin substitutes were produced according to the partially modified self-assembly technique, a method used to reconstruct tissues free of exogenous materials using only donors’ cells [27]. The production of T cell-enriched skin substitutes has been described previously [4]. Briefly, fibroblasts were cultured for 28 days at 37 °C in the presence of ascorbic acid (50 µg/mL; Sigma-Aldrich, Oakville, ON, Canada) until they formed manipulable dermal sheets. At day 28, the previously polarized Th1 and Th17 cells (see Section 2.3.) were seeded to a total of 0.5 × 10^6^ T cells per sheet on half of the dermal fibroblast sheets, with a ratio of 1:1. At the same time, keratinocytes were seeded on the other dermal fibroblast sheets. Keratinocytes and T cells were individually maintained in submerged culture in their respective medium for one additional week (dermal sheets with keratinocytes were cultivated containing the keratinocyte media and dermal sheets containing T cells were cultivated in the fibroblast media supplemented with 10 U/mL IL-2 and 20 ng/mL IL-23). Moreover, during this additional week, culture media for the dermal sheets containing non-adherent T cells was collected to harvest these cells by centrifugation and to resuspend them in fresh medium at each media change. At day 35, a fibroblast sheet with keratinocytes was superimposed on a T cell-containing fibroblast sheet, thus forming a skin substitute comprising a T cell-enriched dermis and an epidermis. This technique is favorable to T cell retention in the skin substitutes as they are still detectable after 4 weeks of culture [4,11]. These newly formed skin substitutes were then raised to the air–liquid interface and kept in culture for a final 3 weeks in the keratinocyte culture medium depleted of EGF to allow the complete differentiation of the epidermis and supplemented with IL-2 (10 U/mL) and IL-23 (20 ng/mL) until the end of the culture. Culture media were also collected to resuspend T cells in fresh medium at each media change as for the previous week. When indicated, an IL-17A-neutralizing monoclonal antibody (0.1 μg/mL, MT504, MabTech, Cincinnati, OH, USA) was used to validate that the model responded well to a known psoriasis treatment. Furthermore, a ribosomal s6 kinase (RSK) inhibitor (BI-D1870, StemCell Technologies) was also tested to evaluate its effect on the T cell-enriched psoriatic skin substitutes. The monoclonal antibody and RSK inhibitor concentrations used were selected based on MTS assays conducted on primary cultures of human psoriatic keratinocytes. Briefly, primary human psoriatic keratinocytes (5 × 10^4^) were seeded on a human feeder layer (2 × 10^4^) and incubated at 37 °C for 2 days. Three doses of the IL-17A antibody were tested (0.5, 1 and 2 μg/mL) as well as five doses of the RSK inhibitor (1, 5, 10, 20 and 50 μM). Cell viability was then assessed after 24 h by MTS colorimetric assays (Abcam, Waltham, MA, USA). Briefly, 20 μL of MTS was added to each well, and the plates were incubated at 37 °C for 1 h. The control condition for the monoclonal antibody (0 μg/mL) contained only the base medium, whereas for the RSK inhibitor, the control condition (0 μM) contained medium with an amount of DMSO corresponding to that used in the tested concentrations of inhibitor. The optical density was then measured using a microplate reader at a wavelength of 490 nm (Appendix A).

### 2.5. Histology and Immunofluorescence Analyses

Biopsy specimens were fixed with a Histochoice solution (Amresco, Solon, OH, USA) and embedded in paraffin. Five-micrometres-thick microtome sections were stained with hematoxylin and eosin for histological analysis. For immunofluorescence analyses, biopsy specimens were embedded in optimal cutting temperature compound (Somagen, Edmonton, AB, Canada), quick frozen in liquid nitrogen and stored at −80 °C until use. Cryosections (6 µm thick) were fixed with cold 100% acetone (10 min at −20 °C) and then incubated for 45 min with the primary antibody directed against Ki-67 (See Appendix A for a detailed description of the antibodies used). Samples were washed with PBS and incubated for 30 min in a dark chamber with the Alexa Fluor 488-conjugated anti-mouse secondary antibody (Appendix A). All antibodies were diluted in PBS containing 1% bovine serum albumin. The slides were assembled using a mounting medium containing 4′-6-diamidino-2-phenylindole (DAPI) Fluoromount-G (SouthernBiotech, AL, USA). Finally, tissue samples were observed with an epifluorescence microscope (Zeiss Canada Ltd., North York, ON, Canada) with an AxioCam HR Rev3 camera (Oberkochen, Germany).

### 2.6. Gene Expression Profiling

All microarray analyses were conducted by the CUO-Recherche gene profiling service (Québec, QC, Canada), as previously described, from the 3 healthy and 4 psoriatic populations mentioned in Section 2.1. [7,28]. The epidermis and dermis were separated mechanically using forceps and a scalpel. It should be noted that at the time of separation, the epidermis had already matured for 28 days on the dermal sheets, which included 7 days of amplification followed by 3 weeks of air–liquid culture in order to allow its complete vertical stratification. Tissues were quick frozen in liquid nitrogen and kept at −80 °C until use. Total RNA was then isolated from the epidermal fractions using the RNeasy Mini Kit (QUIAGEN, Toronto, ON, Canada). The quantity and quality of total RNA were assessed using an Agilent Technologies 2100 bioanalyzer and RNA 6000 Nano LabChip kit (Agilent Technologies, Mississauga, ON, Canada). Cyanine 3-CTP labeled cRNA were prepared from 100 ng of total RNA using the Agilent One-Color Microarray-Based Gene Expression Analysis kit (Agilent Technologies). Then, 600 ng of this freshly prepared cRNA was incubated on a G4858A SurePrint G3 Human GE 8 × 60 K array slide (60,000 probes, Agilent Technologies). Slides were hybridized, washed, and scanned on an Agilent SureScan Scanner according to the manufacturer’s protocol. The transcriptomic results obtained therefore represent the conditions at the end of the skin substitute production, thus after 28 days of epidermal maturation. All data generated from the arrays were analysed by robust multi-array analysis (RMA) for background correction of the raw values. They were then transformed in Log_2_ base and quantile normalized before a linear model was fitted to the normalized data to obtain an expression measure for each probe set on each array. Scatter plots and heat maps were generated using the ArrayStar V12 (DNASTAR, Madison, WI, USA) software. All microarray data presented in this study comply with the Minimum Information About a Microarray Experiment (MIAME) requirement. The gene expression data have been deposited in NCBIs Gene Expression Omnibus (GEO) and are accessible through GEO Series accession number GSE206311 (https://www.ncbi.nlm.nih.gov/geo/query/acc.cgi?acc=GSE206311, accessed on 17 June 2021).

### 2.7. Bioinformatics and Statistical Analyses

The ArrayStar microarray linear expression data for all seven epidermal fractions (3 healthy and 4 psoriatic) were uploaded into the Network Analyst (https://www.networkanalyst.ca/, accessed on 22 October 2021), a web tool based on R language. Data were normalized using the variance stabilizing normalization method, and filtered to exclude low abundance (5th percentile) and low variance (15th percentile) genes. A PCA analysis was subsequently carried out with Network Analyst to determine how populations clustered. A pairwise differential gene expression analysis was then carried out between short and long PMI populations using the limma statistical method, which resulted in a list of statistically differentially expressed genes (adjusted *p*-value < 0.05). This list was then uploaded to and analysed with the Ingenuity Pathway Analysis (IPA, QIAGEN Inc., Venlo, The Netherlands) software to compute and visualize gene interaction networks built around cellular functions of interest (https://www.qiagenbioinformatics.com/products/ingenuitypathway-analysis, accessed on 22 October 2021) [29]. The maximum number of nodes from the functions of interest was set to three to limit the size of the generated interactomes, and the in silico prediction tools of IPA were used to examine how the differentially expressed genes would affect these functions.

### 2.8. Western Blot Analysis

For analyses made from 3D reconstructed skin substitutes, the epidermis was separated from the dermis using forceps. Samples were quick frozen in liquid nitrogen and kept at −80 °C until analysis. To improve the total protein extraction, the tissues were ground. To this end, samples were placed in a safe-lock 2.0 mL Eppendorf tube (ATS Scientific, Burlington, ON, Canada) with two 5 mm stainless beads and ground to a fine powder using a Cryomill MM400 (Retsch, Newton, PA, USA). Total proteins were then extracted by adding 500 μL of RIPA buffer (25 mM Tris-HCL, pH 7.6, 150 mM NaCl, 1% NP-40, 1% sodium deoxycholate, 0.1% sodium dodecyl sulfate (SDS)) containing protease inhibitor Complet™ (Roche, Mannheim, Germany) to each epidermal sample. Samples were incubated on ice for 10 min and centrifuged at 12,000× *g* for 20 min at 4 °C. For the analyses made from the monolayer co-culture (See Section 2.10), cells were detached from the culture flasks using trypsin and harvested to proceed immediately to the extraction of nuclear proteins as described [30]. Samples were quantified using a Pierce BCA Proteins Assay kit (Thermo Fisher Scientific, Rockford, IL, USA), and equal amounts of protein (5 μg for NF-κB, Sp1 and Actin, 15 μg for phospho-Sp1 (p-Sp1) and 30 μg for phospho-NF-κB (p-NF-κB)) were deposited and separated on a 10% reducing SDS-PAGE gel and transferred onto an Immuno-Blot PVDF membrane (Bio-Rad Laboratories, Mississauga, Canada) at 25 volts overnight. Membranes were blocked for 1 h in Tris-buffered saline 0.1% Tween-10 and 5% nonfat milk. Blots were then incubated either overnight (PTPμ, NELL2, p-Sp1, Sp1, p-NF-κB, NF-κB) or for 1 h (ERK1/2, phospho-ERK1/2 (pERK1/2), actin-β) with the primary antibodies and 1 h with the secondary antibodies (see Appendix A for a detailed description of the antibodies used). Proteins were detected using ECL Prime Western Blotting Detection Reagent (GE Healthcare, Little Chalfont, UK) and the Fusion Fx7 imager (Montreal Biotech Inc, Quebec, QC, Canada). Finally, quantification of the immunoblot was performed by densitometry using ImageJ software (National Institutes of Health, Bethesda, MD, USA). Quantification values correspond to the ratio of the protein of interest signal over that of actin.

### 2.9. Enzyme-Linked Immunosorbent Assay (ELISA)

Culture media were collected on days 1, 14 and 21 of the air–liquid interface culture for both healthy and psoriatic conditions. The secretion of IFNκ was evaluated using an IFNκ Human ELISA kit (MBS936153; MyBioSource, San Diego, CA, USA). All procedures were conducted at room temperature, and samples were assayed in duplicate. The reaction was measured at 450 nm using a SpectraMax Plus 384 microplate spectrophotometer (Molecular Devices, San Jose, CA, USA).

### 2.10. Electrophoretic Mobility-Shift Assay

Healthy (46 and 49 years old) and psoriatic (46 and 49 years old) keratinocytes (passage 3) were first cultured for six days at 37 °C on a human feeder layer (8 × 10^3^ cells per cm^2^) until they reached near-confluence (about 90% coverage of the culture flasks). On the last day of culture, keratinocytes were stimulated for 24 h with T cells (6.7 × 10^3^ cells per cm^2^) and isolated, activated and polarized as described in Section 2.3. Subsequently, the cells were detached from the culture flasks using trypsin and harvested to proceed immediately to the extraction of nuclear proteins as described in [30]. Protein concentration was evaluated by the Bradford procedure. EMSAs were conducted as previously described by incubating nuclear extracts with a 5′-^32^P-end-labeled double-stranded oligonucleotide bearing the high-affinity binding site for either Sp1/Sp3 or NF-κB (See Appendix A for DNA sequences used) [30]. Briefly, a 8 × 10^4^ cpm labeled probe was incubated with 10 μg of crude nuclear proteins prior to separation of the DNA–protein complexes by EMSA. Competition experiments were performed by adding unlabeled double-stranded oligonucleotides (50- to 500-fold molar excesses) bearing the binding site for the transcription factors Sp1/Sp3, NF-κB or nuclear factor I (NFI) (See Appendix A for DNA sequences used). Supershift experiments were conducted by incubating nuclear extracts in the presence of 1 μg antibodies against Sp1 (ab227383, Abcam, Toronto, ON, Canada), Sp3 (sc365220, Santa Cruz Biotechnology, Dallas, TX, USA) or NF-κB (sc8008, Santa Cruz Biotechnology, Dallas, TX, USA). Formation of the DNA–protein complexes was then revealed by gel electrophoresis through 8% native polyacrylamide gels run 6H in Tris-glycine buffer (50 nm Tris, 2.5 mM EDTA, 0.4 M glycine) at 4 °C. The gels were dried and autoradiographed overnight at −80 °C.

### 2.11. Statistical Analysis

Data are presented as mean ± standard deviation. Statistical analyses were performed using ANOVAs followed by Tukey’s post hoc test, except when stated otherwise. Only values of *p* < 0.05 were considered significant. All calculations were performed with Prism software, version 5 (GraphPad Software, San Diego, CA, USA)

## 3. Results

### 3.1. Skin Substitute Morphology

Human-tissue-engineered skin substitutes were produced according to the self-assembly method using cells from patients with either psoriasis or healthy skin. The results show that the psoriatic phenotype was retained in the T cell-enriched psoriatic skin substitutes (PS^+T^) compared with healthy control skin substitutes (HS^+T^). For instance, the macroscopic aspect of PS^+T^ exhibited a less uniform appearance with more relief compared with HS^+T^ (Figure 1a,b). In addition, keratinocyte hyperproliferation, a key feature of psoriasis, is manifested by the increased thickness of the living epidermis of PS^+T^ compared with HS^+T^ (Figure 1e,f,m), as well as the increased number of Ki-67-positive cells (Figure 1i,j,n). Two treatments were also performed on psoriatic substitutes. First, an IL-17A-neutralizing antibody was used to validate that the PS^+T^ model responded appropriately to a known psoriasis treatment (PS^+T+IL-17A mAb^). More detailed analyses of anti-IL17a-neutralizing antibody efficacy have been presented elsewhere [4]. BI-D1870, a ribosomal s6 kinase (RSK) inhibitor, was also used to investigate the impact of inhibiting a target of the ERK1/2 signaling pathway and to evaluate its potential on PS^+T^ (PS^+T+RSK inhibitor^). Both treatments restored the phenotype closer to that observed in the healthy skin by decreasing the proliferation of psoriatic keratinocytes. Indeed, based on histological analyses, both PS^+T+IL-17 mAb^ and PS^+T+RSK inhibitor^ displayed decreased acanthosis. Measurements confirmed that both treatments reduced epidermal thickness (Figure 1f–h,m). Moreover, the detection of the proliferation marker Ki-67 by immunofluorescence revealed that both treatments restored the proportion of Ki-67-positive cells to that of healthy controls (Figure 1j–l,n).

### 3.2. Differentially Expressed Genes between Healthy and Psoriatic Skin Substitutes

Gene profiling analyses were conducted using total RNA extracted from T cell-enriched skin substitutes produced with cells either from three different healthy donors (HS^+T^; 18, 46 and 49 years old) or from four different psoriatic patients (PS^+T^; 36, 46, 49 and 64 years old) and used for microarray analyses. The scatter plot analysis of the probes loaded on the chip showed important changes in the pattern of genes expressed by PS^+T^ compared with HS^+T^ (Figure 2a, R2 = 0.976). Analysis of the different transcripts contained on the microarray revealed that 1549 genes were differentially expressed by more than two-fold between HS^+T^ and PS^+T^, including 595 genes deregulated by more than three-fold. The results from this analysis were then uploaded into the Ingenuity Pathway Analysis (IPA) software to establish an interaction network among these differentially expressed genes for biological functions of interest, namely, proliferation, differentiation and migration (Figure 2b). As shown on the heatmap presented in Figure 2b, the 10 most significantly (*p* ˂ 0.05) altered genes identified through the IPA analysis were also identified among the most differentially expressed by gene microarray (Figure 2b). Among these, four genes showed decreased expression (*PTPRM*, *NR2F2*, *KRT31*, *IFNK*) in PS^+T^ while the other six genes (*NELL2*, *IRX1*, *CCDC8*, *POU2AF1*, *MCHR1*, *TRPS1*) were rather overexpressed. Moreover, the deregulated genes identified in this study were also compared to three transcriptomic studies performed from psoriatic skin biopsies [3,31,32]. Of the 60 randomly selected genes among those identified as deregulated in psoriasis by Gudjonsson et al., Li et al. and Pasquali et al., a total of 38 (63%) were similarly deregulated between PS^+T^ and HS^+T^ conditions and a total of 49 (82%) were deregulated between PS^+T^ and HS^−T^ conditions (Appendix A).

### 3.3. Validation of Deregulated Genes at the Proteomic Level

The transcriptomic results were further validated at the proteomic level by performing Western blots against PTPμ (encoded by the *PTPRM* gene) and NELL2 proteins, as well as through an ELISA assay for IFNκ secretion. Consistent with the gene expression profile on the microarray, PTPμ expression is decreased in PS^+T^ compared with HS^+T^, while the opposite is observed for NELL2, with an increase in its expression in PS^+T^ compared with HS^+T^ (compare Figure 3a and Figure 2c). In addition, healthy populations 1 and 3 showed higher levels of PTPμ compared with healthy population 2. The same was also observed for the psoriatic populations 1, 2 and 4, which exhibited higher levels of expression of NELL2 than psoriatic population 3. On the other hand, the levels of secreted IFNκ are in contradiction with the transcriptomic data (compare Figure 2b and Figure 3d). However, one should keep in mind that the substitutes used for the transcriptomic analyses were those at the end of the cell culture, i.e., at day 21 of air–liquid culture. Consistent with the transcriptomic data, IFNκ could not be detected in the culture supernatants on day 21 of air–liquid culture for both healthy and psoriatic substitutes (Figure 3d). However, important levels were detected on day 1 and 14 of air–liquid culture in the supernatants of the PS^+T^, particularly on day 1, where the amount was significantly higher for the psoriatic condition compared with the healthy one (Figure 3d). The increased production of IL-17A and IFNγ in the supernatant of PS^+T^ was also validated, confirming the evidence of the pro-inflammatory microenvironment specific of psoriasis (Figure 3e,f).

### 3.4. Contribution of NELL2 and PTPμ to the ERK1/2 Signaling Pathway in Psoriasis

We next conducted Western blot analyses in order to determine whether the IL-17A-neutralizing antibody or the RSK inhibitor could affect the activation of the ERK signaling pathway in our psoriatic skin substitutes. As expected, ERK1/2 phosphorylation is greater in PS^+T^ than in HS^+T^ (3.2-fold increase; Figure 4a,b). However, the addition of the IL-17A-neutralizing antibody considerably decreased the phosphorylation levels of ERK1/2 (4-fold reduction; Figure 4a,b). RSK is a downstream target of ERK1/2, and its inhibition is therefore not expected to have any impact on ERK1/2 activity. As expected, the inhibition of RSK had no impact on the level of phosphorylated ERK1/2 (Figure 4a,b).

### 3.5. DNA-Binding Properties of Sp1 and NF-κB in Healthy and Psoriatic Keratinocytes Stimulated by T Cells

Transcription factors mediate many cellular processes by initiating and regulating gene transcription. They can act as both activators and/or repressors of transcription. In order to investigate in more detail the differences in ERK1/2 phosphorylation between healthy and psoriatic conditions, we evaluated the DNA-binding properties of two transcription factors that are downstream nuclear targets of ERK1/2, namely, Sp1 and NF-κB, in nuclear extracts prepared from monolayers of either healthy or psoriatic keratinocytes stimulated with T cells. EMSA analyses clearly demonstrated that the DNA-binding properties of these two transcription factors are strongly increased in psoriatic keratinocytes compared with healthy ones (Figure 5a,b). The specificity of the complex formation was validated by competition experiments. Indeed, the formation of both the Sp1 and NF-kB complexes was prevented only by unlabeled competitor oligonucleotides bearing their respective high-affinity DNA binding sites (Figure 5c,d). Moreover, the addition of antibodies directed against either Sp1 or NF-κB significantly reduced the formation of these complexes in EMSA and also produced new, slow-migrating supershifted complexes (SCC), confirming once again the presence of both Sp1 and NF-κB in these DNA–protein complexes (compare lanes 7 and 8 with lane 2 (negative control) for Sp1 and Sp1/Sp3 in Figure 5c and lane 7 with lane 2 (negative control) for NF-κB on Figure 5d). We then evaluated the expression of both Sp1 and NF-κB by Western blot and gene profiling analyses to establish whether the increased DNA binding observed for these transcription factors in the EMSA resulted from a corresponding increase at the protein level or from an increased phosphorylation of these factors (Figure 5g–j). Interestingly, both the protein and transcription levels remained very much the same for Sp1 and NFkB in both the healthy and psoriatic conditions (Figure 5e–j). Therefore, these results suggest that the elevated DNA-binding activity observed for these transcription factors in the psoriatic condition clearly results from their increased phosphorylation through the activation of the ERK1/2 pathway and not from a corresponding increase in their respective protein concentrations.

## 4. Discussion

Psoriasis is a complex immune-mediated skin disease characterized mainly by keratinocyte hyperproliferation. Although the signalization involved in the pathogenesis of psoriasis has been investigated in recent years, much remains to be discovered. In this regard, our previous studies aimed to investigate the cellular and molecular mechanisms of psoriasis [7,8,10,13,33,34,35,36]. More recently, we developed a more complex biomaterial allowing the study of the interaction between skin and immune cells in a microenvironment that closely mimics that of the pathology [4]. In this study, the molecular pathogenesis of psoriasis was evaluated using both transcriptomic and proteomic approaches on our new T cell-enriched psoriatic skin biomaterial. We showed that two genes of interest, namely, *PTPRM* and *NELL2*, were deregulated in psoriatic skin substitutes and led to the overactivation of the ERK1/2 signaling pathway (Figure 6). Furthermore, our results show that this excessive activation of ERK1/2 in the psoriatic condition also increased the DNA-binding properties of the transcription factors NF-κB and Sp1, two downstream targets of the ERK1/2 pathway that play important roles in many different biological functions, including cell proliferation and migration.

Some studies have already reported the excessive activation of the ERK1/2 signaling pathway in psoriasis [20,21,37,38,39,40]. Consistent with this, the inhibition of this signaling pathway by a specific ERK1/2 inhibitor decreased psoriatic lesions in an imiquimod-induced murine psoriasis model [41]. However, the detailed mechanism by which this signaling pathway is overactivated remained to be further investigated. Activation of the ERK1/2 signaling pathway was previously reported in our tissue-engineered PS model without T cells [33]. Here, we have shown that the addition of T cells to our previous model further supports this observed activation and that treatment with an IL-17A-neutralizing antibody reduces ERK1/2 phosphorylation levels. It is reported that the MAPK signaling pathway may be involved in IL-17-mediated inflammation. Indeed, IL-17A is known to signal through several intracellular pathways such as ERK and to promote proliferation [42,43]. Our results are in agreement with previous studies showing that IL-17 induced ERK1/2 activation [44,45,46,47,48]. IL-17A-neutralizing antibodies are well established as a treatment of psoriasis [49,50,51]. By performing this treatment on PS^+T^, we aimed to validate that this T cell-enriched model was responsive to an already-known treatment for psoriasis, thus making it an interesting tool for the development of new therapeutic targets. In this sense, the clear contribution of the ERK1/2 signaling pathway in psoriasis led us to target one of its downstream effectors, the ribosomal s6 kinase (RSK), and to evaluate the effect of its inhibition on the PS^+T^ morphology. Our findings show that the inhibition of rSK in PS^+T^ possibly restored characteristics that are much closer to those of healthy skin by reducing keratinocyte hyperproliferation. RSK is reportedly involved in numerous skin cancers by inducing cell cycle progression, cell proliferation and anchorage-independent cell transformation [52]. Since psoriatic keratinocytes are also characterized by excessive proliferation and anti-apoptotic properties, the inhibition of RSK in PS^+T^ is consistent with the observed decrease in keratinocyte proliferation [53,54].

As previously mentioned, our transcriptomic data identified two genes of interest that can potentially influence the ERK1/2 signaling pathway, namely, *PTPRM* and *NELL2*. First, *PTPRM* is deregulated in PS^+T^ compared with HS^+T^, both at the transcript and proteomic levels. *PTPRM* encodes for the protein tyrosine phosphatase (PTPs) receptor type M (PTPμ). PTPs are known to be signaling molecules that regulate a variety of cellular processes including cell growth, differentiation, apoptosis and cell movement [55]. Some studies have revealed a decrease in the expression of this gene in psoriatic skin, without, however, providing an explanation as to its contribution to the pathology [3,32,56]. The dynamic balance of cellular phosphotyrosine levels is achieved by the opposing action of protein tyrosine kinases and protein tyrosine phosphatases. PTPμ has a structure similar to that of cell adhesion molecules. In addition, it has been shown to exhibit homophilic binding and to confer cell–cell adhesion in epithelial and cancer cells [57,58]. Sun and colleagues were the first to identify a correlation between decreased *PTPRM* expression and poor prognosis in breast cancer. Moreover, patients with low *PTPRM* expression had shorter survival times than those with higher levels of *PTPRM* levels [59]. Of note, their findings revealed that the knockdown of *PTPRM* resulted in elevated adhesion, invasion and proliferation of breast cancer cells through the regulation of the tyrosine phosphorylation of ERK. These results are consistent with our data indicating a decrease in *PTPRM* expression in PS^+T^ that also results in an enhanced level of ERK phosphorylation. In another study, Sui and colleagues reported that *PTPRM* knockdown in human lung microvascular endothelial cells impaired the endothelial barrier function, whereas its overexpression improved it [60]. PTPμ interacts with vascular endothelial (VE)-cadherin and regulates both the tyrosine phosphorylation state of VE-cadherin and the integrity of the barrier function [60]. In a similar perspective, these results can be extrapolated to psoriatic skin, which is also characterized by an impaired epidermal barrier function [61,62,63]. PTPμ is also known to interact with other classical cadherins, including epithelial (E-) cadherin [64]. In line with this hypothesis, Chung and colleagues reported that total E-cadherin levels are dramatically reduced in psoriatic lesions in a murine skin model treated with amphiregulin, a keratinocyte growth factor overexpressed in psoriasis. Moreover, E-cadherin downregulation was detected in the upper granular layer, as well as in the basal layer in human psoriatic tissue [65]. The integration of these interesting results with the findings of our study suggests for the first time a contribution of *PTPRM* to the pathogenesis of psoriasis.

*NELL2* is another gene whose expression was deregulated in PS^+T^ compared with HS^+T^. Indeed, both the NELL2 transcript and protein levels are increased in our PS^+T^ model. *NELL2* codes for the neural epidermal growth factor-like 2. Neural growth factors are produced by several cell types, including keratinocytes and fibroblasts. They promote nerve growth and increase neuropeptide regulation [66]. NELL2 has been shown to promote neuronal polarization and axon growth through the ERK signaling pathway. Consistent with these results, our PS^+T^ model has an increased expression of NELL2 compared to HS^+T^, which also results in increased activation of the ERK1/2 signaling pathway in PS^+T^. Little is known regarding the increased expression of *NELL2* in psoriatic skin. However, the expression of this gene is known to be increased in other skin diseases such as atopic dermatitis [6]. It is possible that the simplified environment of the model used compared to the complexity of native psoriatic skin may reveal some differences. More studies are needed to establish whether *NELL2* might contribute to the pathogenesis of psoriasis.

A potential association between the increase in *NELL2* expression in psoriatic skin and the increased number of nerve fibers found in the psoriatic epidermis may be an interesting hypothesis. The nervous system is reported to play an essential role in the development of psoriasis [67]. Indeed, various clinical reports show a spontaneous improvement in psoriatic lesions after central or peripheral nerve damage [68,69,70]. Moreover, in some cases, the return of neurological function was also followed by a return of the skin lesions [71]. Related to the above, in a study that included twenty-four patients with psoriasis, Taneda and colleagues reported that the number of epidermal nerve fibers tended to increase in approximately 40% of psoriatic patients experiencing itching compared with healthy controls [72]. Further research will need to be undertaken to establish the contribution of NELL2 in psoriasis, but the overproduction of NELL2 in PS^+T^ may offer an explanation for the hyper-innervation that is found in psoriasis.

Another interesting finding from the transcriptomic data is related to *IFNK* expression. *IFNK* encodes interferon kappa (IFNκ), a type I interferon expressed by keratinocytes that has been reported both to be increased and decreased in psoriatic lesions, in addition to being a major source of type I IFN in the healthy control epidermis [73,74,75,76]. The deletion of the IFN-1 receptor has been reported as being protective in an imiquimod-induced murine psoriasis model [77]. Using this same murine model, another report suggests that the absence of type I IFN signaling makes no difference in the disease phenotype [78]. In this study, the *IFNK* transcript was found to be overexpressed in HS^+T^ compared with PS^+T^. It is important to note that the microarray experiments were only performed on the last day of culture; thus, after 21 days of air–liquid culture, when the epidermis has fully matured. On the other hand, the ELISA assay allowed us to follow the production of IFNκ over time (from 1 to 21 days of culture). Interestingly, the amount of IFNκ produced was higher in PS^+T^ than in HS^+T^ in the first two weeks of air–liquid culture (at both day 1 and day 14), but then had completely disappeared in both PS^+T^ and HS^+T^ at day 21, a result also consistent with our transcriptomic data. Our findings support other studies reporting the detection of a type I IFN signature that occurs early in the development of psoriatic lesions, suggesting a role for type I IFN in the initiation of inflammation in psoriasis [74,79,80]. Moreover, as reported by Gharaee–Kermani and colleagues, the timing and location of biopsies may affect these results, thus explaining the difference obtained between our transcriptomic results at the end of culture and the amounts of IFNκ previously produced, as well as the discrepancy in IFNκ production between different studies [74]. Furthermore, type I IFN is also known to activate the ERK signaling pathway [81]. Although the quantity of IFNκ on the last day of culture was below the detection threshold, the activation of the ERK1/2 pathway was still observed in PS^+T^, suggesting that ERK1/2 activation results from multiple signals, many of which are still functional after 21 days of air–liquid culture.

These data were obtained with a complex model using pathological skin cells, but enriched with activated T cells from healthy donors, thus constituting possibly the main limitation of our model. Indeed, the development of completely autologous skin models represents a major challenge. One of the reasons behind this difficulty is the cryopreservation of T cells. Fibroblasts and keratinocytes are isolated from psoriatic lesions and properly frozen to generate a psoriatic skin cell bank. These cell banks can then be generated over several years, and the skin cells are thawed to produce skin substitutes when needed. However, the same is not true for T cells since the impact of freezing is not yet validated, thus preventing the production of a relevant T cell bank matching that of psoriatic skin cells [82]. It should be noted that, to our knowledge, there is no 3D in vitro model produced using psoriatic skin cells and psoriatic T cells. The presence of complex models, either with a combination of psoriatic skin cells and healthy T cells as in this study or with a combination of healthy skin cells and psoriatic T cells as developed by Shin et al., remains very interesting and provides complementary and relevant information regarding the study of the molecular and cellular mechanisms of psoriasis [4,5].

The exact comparison between transcriptomic studies performed on native psoriatic skin and our model is challenging considering the differences between these types of studies. However, taking these differences into consideration, it is still possible to raise some relevant conclusions. Among in vivo transcriptomic studies performed on native psoriatic skin, Gudjonsson and coworkers identified an impressive number of deregulated genes between healthy and psoriatic skin biopsies using a microarray [31]. In addition, using RNA-seq, Li and coworkers highlighted the contribution of transcripts with low levels of expression which could not be identified by microarray studies [32]. In addition, this study raised the deregulation of four of the top deregulated genes identified in PS^+T^, namely, *PTPRM*, *MCHR1*, *KRT31* and *NR2F2*, in psoriatic skin which had not been identified by the study of Gudjonsson and coworkers. Along with the findings in the study by Li et al., the downregulation of *KRT31* is highlighted in other studies investigating its expression from epidermal keratinocytes directly instead of global skin analysis [3,7,83]. Pasquali and colleagues also performed a relevant study by evaluating the transcriptomic profile of keratinocytes extracted and isolated from healthy and psoriatic skin biopsies through a selection of CD45-negative epidermal cells [3]. From the 60 randomly selected genes from those identified as deregulated in psoriasis by Gudjonsson et al., Li et al. and Pasquali et al., a total of 39 (65%) were similarly deregulated between PS^+T^ and HS^+T^. Interestingly, the study performed by Pasquali et al. was the only one of these three in which keratinocytes were isolated from biopsies prior to transcriptome analysis. Consequently, this study allowed to highlight for the first time the deregulation of the *TRPS1* gene, coding for a transcription factor involved in the regulation of epidermal differentiation and inflammation [3]. This study suggests an improvement in the gene profile compared to our previous T cell-free psoriatic skin model that showed 54% similarity (Appendix A) [7]. These data are encouraging, especially since this new T cell-enriched psoriatic skin model still lacks some components, such as the presence of nerves and other immune cell types. Moreover, in contrast to healthy skin which is not considered to be in an inflammatory context, native psoriatic skin is in a state of chronic inflammation where multiple immune cells are recruited and produce various pro-inflammatory cytokines. Considering this, the PS^+T^ model was also compared to the HS^−T^ one to reflect this condition (Appendix A). This comparison revealed 50 genes out of 60 (83%) that were similarly deregulated with the two in vivo transcriptomic studies, as well as raising important differences that were attenuated when compared to HS^+T^ (Appendix A). Since the transcriptomic profile obtained in this study is mainly from epidermal keratinocytes, we believe that comparing PS^+T^ to HS^−T^ is similar to comparing keratinocytes from native psoriatic skin in an inflammatory state to those from healthy skin. Beyond the differences caused by the composition of our skin model compared to native skin, it is relevant to point out that RNA isolated directly from biopsies does not go through the same processes as those isolated from our skin substitutes. These differences may influence the gene expression profile and explain the additional discrepancies observed between these two types of studies.

Finally, two downstream nuclear targets of ERK1/2 were also investigated, namely, the transcription factors NF-κB and Sp1. NF-κB is one of the most potent activating transcription factors of proinflammatory gene activity, thereby making it an important mediator of inflammation in the pathogenesis of psoriasis [84]. As for Sp1, it is a well-known, ubiquitous transcription factor that participates in the regulation of tissue-specific response genes, as well as cell cycle and signalling pathways [85,86]. In this study, NF-κB and Sp1 showed increased DNA-binding properties in psoriatic keratinocytes compared with healthy ones without displaying different expression levels at both the transcriptomic and proteomic levels. These results are in agreement with the high levels of NF-κB activation found in the inflamed skin. Indeed, several treatments aimed at decreasing active NF-κB levels have demonstrated antipsoriatic effects, including tumor necrosis factor-α blockers and glucocorticoids [87,88]. Moreover, dimethyl fumarate, a drug used for decades in the treatment of psoriasis, has been identified as an NF-κB inhibitor, specifically targeting the C-terminal kinase domain of RSK [89,90]. Little information is available about the involvement of Sp1 in psoriasis. However, the progressive loss of Sp1 expression during higher passages of human skin keratinocytes has been correlated with their growth arrest and terminal differentiation [30,91]. Taking that into account, the increase in Sp1 activity under psoriatic conditions is expected to support the hyperproliferation of keratinocytes and their impaired differentiation in the pathology. The increased DNA-binding properties of both NF-κB and Sp1 are also supported by the excessive activation of ERK1/2 observed in our psoriatic biomaterials.

## 5. Conclusions

In conclusion, this study further investigated the molecular and cellular mechanisms of psoriasis using a novel T cell-enriched psoriatic skin biomaterial. Our findings suggest for the first time a contribution of the *PTPRM* gene product to the pathogenesis of psoriasis. The excessive activation of the ERK1/2 pathway in psoriatic substitutes can be explained by decreased *PTPRM* expression, which promotes keratinocyte survival and proliferation. Because of its involvement in the excessive proliferation of pathological keratinocytes, the PTPµ/ERK1/2 signaling pathway could represent a promising avenue for the treatment of psoriasis. This model provides a valuable tool for the study of disease mechanisms and for the validation of potential therapeutic targets.

## Figures and Tables

**Figure 1 cells-11-02904-f001:**
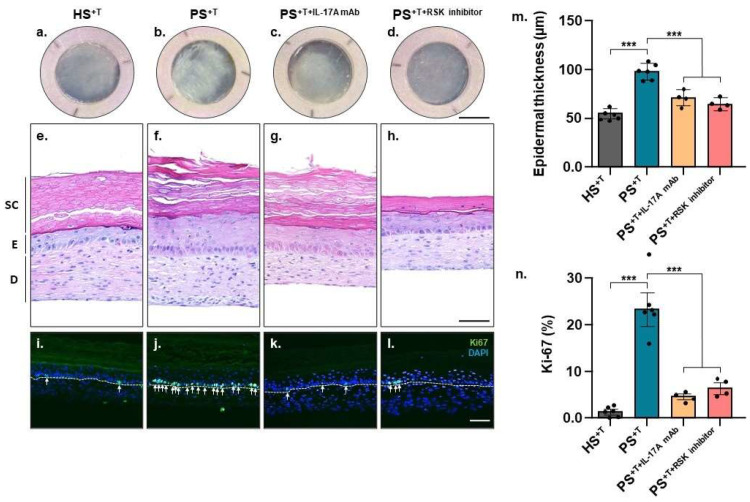
Morphological aspect of the skin substitutes. (**a**–**d**) Macroscopic aspect and (**e**–**h**) histological cross-section following hematoxylin and eosin staining of the skin substitutes. (**i**–**l**) Detection of Ki67-positive cells (in green) by immunofluorescence. Epidermal thickness quantification from hematoxylin and eosin staining (**m**) and the ratio of Ki67-positive cells to the total number of basal keratinocytes (**n**) are illustrated. White arrows highlight Ki67-positive cells. Statistical significance was determined using one-way ANOVA followed by Tukey’s post hoc test, *** *p* < 0.001 (*n* = 3 donors per condition, except for treated conditions (*n* = 2), *n* = 2 skin substitutes per donor). Scale bars: (**a**–**d**) 1 cm, (**e**–**l**) 100 µm. HS^+T^, healthy skin substitutes with T cells; PS^+T^, psoriatic skin substitutes with T cells; PS^+T+IL-17A mAb^, psoriatic skin substitutes with T cells treated with an IL-17A-neutralizing antibody; PS^+T+RSK inhibitor^, psoriatic skin substitutes with T cells treated with a RSK inhibitor; SC, *stratum corneum*; E, epidermis; D, dermis.

**Figure 2 cells-11-02904-f002:**
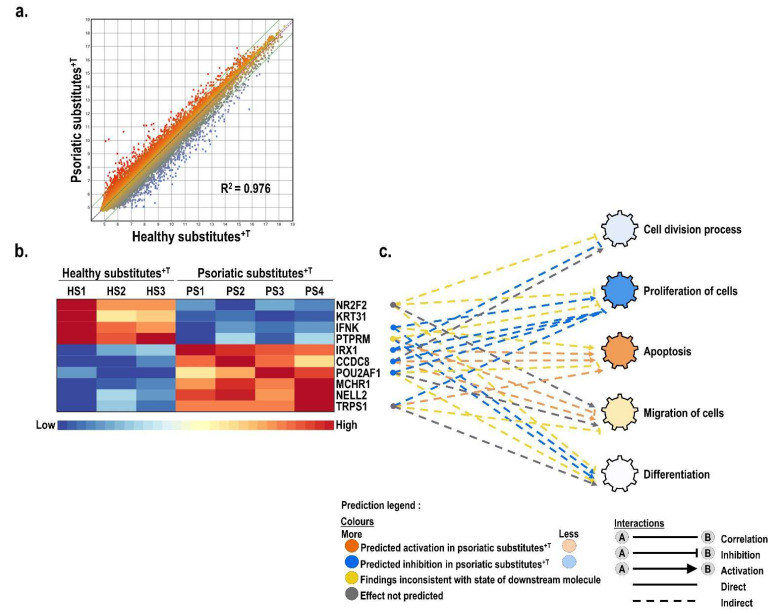
Microarray analysis of the gene expression pattern between T cell-enriched healthy and psoriatic skin substitutes. (**a**) Scatter plots of log2 of signal intensity from 60,000 different probes covering the entire human transcriptome of HS^+T^ (*x*-axis) against PS^+T^ (*y*-axis). The color scale corresponds to the level of gene expression (dark blue being the lowest and dark red the highest) (**b**) Heatmap representation of the expression profile for the 10 genes identified through the IPA analysis as the most strongly deregulated between healthy (*n* = 3) and psoriatic (*n* = 4) skin substitutes enriched with T cells. (**c**) Ingenuity pathway analysis of gene interaction networks altered in PS^+T^ condition built around biological functions of interest: proliferation, differentiation, migration, apoptosis and cell division. HS^+T^, healthy skin substitutes with T cells; PS^+T^, psoriatic skin substitutes with T cells.

**Figure 3 cells-11-02904-f003:**
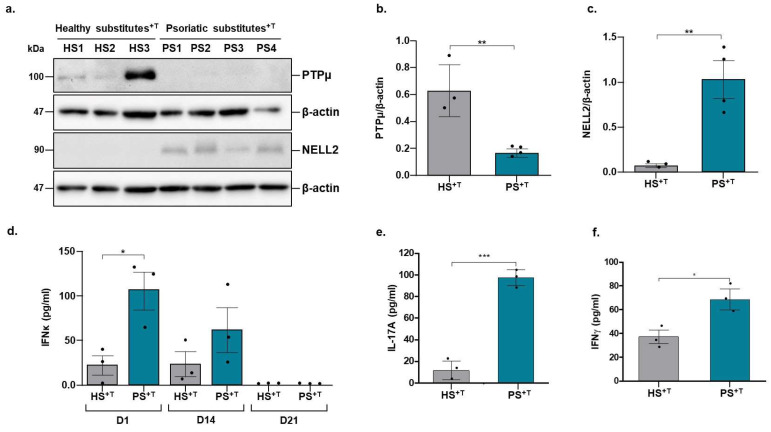
Differential expression of PTPμ, NELL2, IFNκ, IL-17A and IFNƴ between healthy and psoriatic skin substitutes enriched with T cells. (**a**) Expression of PTPμ and NELL2 was monitored by Western blot analyses of total protein extracts prepared from the epidermal fraction of HS^+T^ and PS^+T^ (30 μg and 20 μg total protein, respectively, was loaded on a 10% reducing SDS-PAGE gel). Densitometric analysis of the immunoblots (*n* = 3 healthy donors and 4 psoriatic patients) for the relative expression of (**b**) PTPµ and (**c**) NELL2. Β-actin is shown as a loading control. (**d**) IFNκ production in healthy and psoriatic skin substitutes enriched in T cells was measured by ELISA at days 1, 14 and 21 of air–liquid culture. Increase in IL-17A (**e**) and IFNƴ (**f**) have also been validated at day 1 of air–liquid culture by ELISA assay. The *p*-values were derived from Student’s t-test and considered statistically significant when *p* ˂ 0.05. (* *p* ˂ 0.05, ** *p* ˂ 0.01, *** *p* ˂ 0.001, *n* = 3 donors per conditions). HS^+T^, healthy skin substitutes with T cells; PS^+T^, psoriatic skin substitutes with T cells; D, day.

**Figure 4 cells-11-02904-f004:**
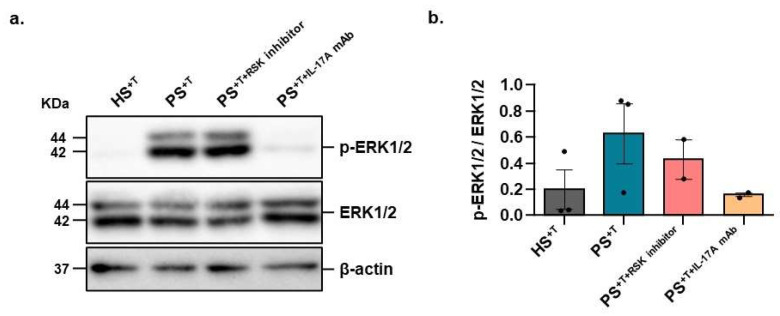
Contribution of ERK1/2 signaling pathway in psoriasis. (**a**) Total protein (20 μg) from the epidermal fraction of HS^+T^, PS^+T^, PS^+T+RSK inhibitor^ and PS^+T+IL-17A mAb^ was loaded on a 10% reducing SDS-PAGE gel. The expression of p-ERK1/2 and total ERK1/2 (ERK1/2) was monitored by Western blot analysis. β-actin is shown as a loading control. (**b**) Densitometric analysis of the immunoblots (three donors (HS^+T^ and PS^+T^) and two donors (PS^+T+RSK inhibitor^ and PS^+T+IL-17A mAb^), two skin substitutes per donor). HS^+T^, healthy skin substitutes with T cells; PS^+T^, psoriatic skin substitutes with T cells; PS^+T+RSK inhibitor^, psoriatic skin substitutes with T cells treated with a RSK inhibitor; PS^+T+IL-17A mAb^, psoriatic skin substitutes with T cells treated with an IL-17A-neutralizing antibody.

**Figure 5 cells-11-02904-f005:**
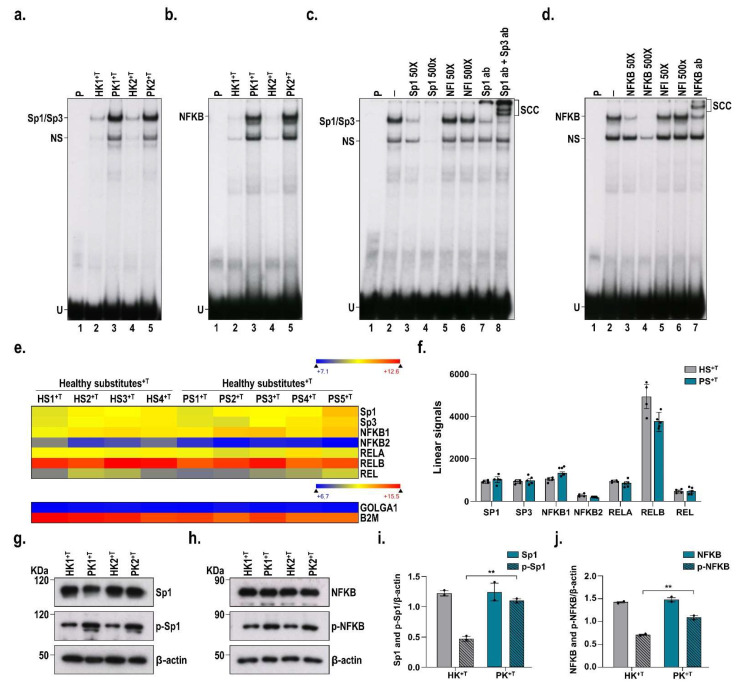
DNA-binding properties of Sp1 and NF-κB in healthy and psoriatic conditions. Electrophoretic mobility shift assays showing the DNA-binding of (**a**) Sp1 and (**b**) NF-κB in nuclear extracts prepared from monolayers of either healthy (HK^+T^) or psoriatic (PK^+T^) keratinocytes stimulated with T cells. When indicated, 50- and 500-fold molar excesses of unlabeled oligonucleotides for Sp1 ((**c**); lines 3 and 4), NF-κB ((**d**); lines 3 and 4) and NFI ((**c**); lines 5 and 6, (**d**); lines 5 and 6) were also added as unlabeled competitors. Antibodies directed against Sp1 and Sp1/Sp3 ((**a**); lines 7 and 8) as well as against NF-κB ((**b**); line 7) allowed the detection of their corresponding supershifted complexes. Heatmap representation (**e**) and associated linear signals (**f**) of the changes in NF-κB and Sp1 gene expression between HS^+T^ and PS^+T^. Western blot analyses for both phosphorylated and non-phosphorylated forms of (**g**) Sp1 and (**h**) NF-κB were performed using nuclear extracts from monolayers of HK^+T^ and PK^+T^. Densitometric analysis of the immunoblots for (**i**) Sp1 and (**j**) NF-κB. Statistical significance was determined using one-way ANOVA followed by Tukey’s post hoc test, ** *p* < 0.05. HK^+T^, healthy keratinocytes stimulated with T cells; PK^+T^, psoriatic keratinocytes stimulated with T cells; HS^+T^, healthy skin substitutes with T cells; PS^+T^, psoriatic skin substitutes with T cells; P, labeled probe alone; U, unbound fraction of the labeled probe; SSC, supershift complex.

**Figure 6 cells-11-02904-f006:**
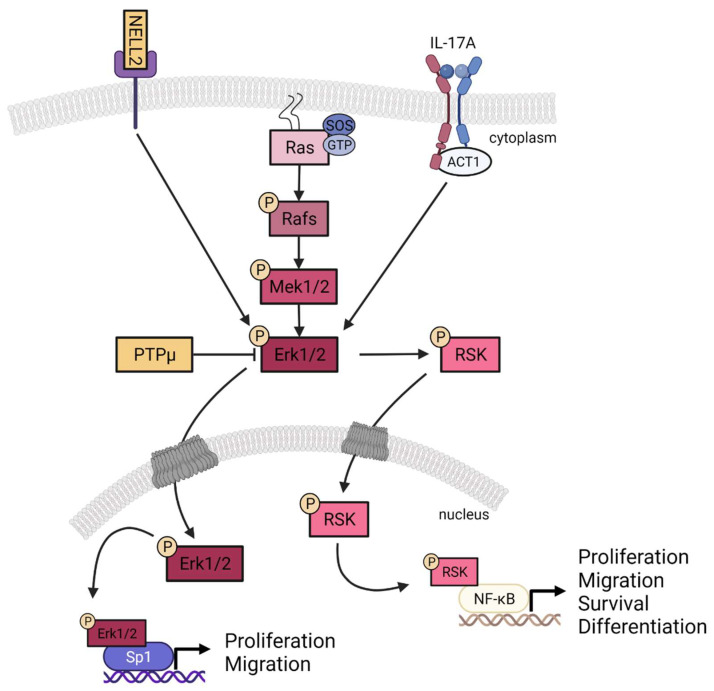
Contribution of PTPμ and NELL2 to the ERK1/2 signaling pathway. NELL2 contributes to the activation of ERK1/2 while PTPμ inhibits its activation through dephosphorylation. In our T cell-enriched psoriatic skin substitutes, NELL2 expression was upregulated, whereas that of PTPμ was decreased, both resulting in an overactivation of the ERK1/2 signaling pathway.

## Data Availability

All microarray data presented in this study comply with the Minimum Information About Microarray Experiment requirements. The gene expression data have been deposited in the National Center for Biotechnology and Information’s Gene Expression Omnibus (http://www.ncbi.nlm.nih.gov/geo/, accessed on 17 June 2022) and are accessible through GEO Series accession number GSE206311 (https://www.ncbi.nlm.nih.gov/geo/query/acc.cgi?acc=GSE206311, accessed on 17 June 2022).

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
