# Peer review of "Gene Profiling of a 3D Psoriatic Skin Model Enriched in T Cells: Downregulation of PTPRM Promotes Keratinocyte Proliferation through Excessive ERK1/2 Signaling"

_cells, 2022, doi:10.3390/cells11182904_

Round 1
Reviewer 1 Report
Psoriasis is a complex immune-mediated skin disease characterized mainly by keratinocyte hyperproliferation. In the study, the molecular pathogenesis of psoriasis was evaluated using both transcriptomic and proteomic approaches on the new T cell-enriched psoriatic skin biomaterial. They showed that two genes of interest, namely PTPRM and NELL2, were deregulated in psoriatic skin substitutes, and led to overactivation of the ERK1/2 signaling pathway. The results show that this excessive activation of ERK1/2 in the psoriatic condition also increased the DNA-binding properties of the transcription factors NF-κB and Sp1, two downstream targets of the ERK1/2 pathway that play important roles in many different biological functions, including cell proliferation and migration. Because of its involvement in the excessive proliferation of pathological keratinocytes, the PTPμ/ERK1/2 signaling pathway could represent a promising avenue for the treatment of psoriasis.
The manuscript is well-structured and clear presented in good English with appropriate graphics. The cited references are mostly recent publications (within the last 5 years) and relevant to the topics discussed in the article. The conclusions is consistent with the evidence and arguments presented base on this complex biomaterial that closely mimics that of psoriasis skin is new and original.
Although the signalization involved in the pathogenesis of psoriasis has been investigated in recent years, much remains to be discovered.
I recommend the paper to be accepted without any changes.
Author Response
We would like to sincerely thank the reviewer for his/her attentive reading of our manuscript and kind comments.
Reviewer 2 Report
The manuscript by Rioux et al reports an interesting study performed in a 3D psoriatic skin model enriched in T cells in which the authors identified two dysregulated genes altering keratinocyte proliferation and ERK signaling (namely PTPRM and NELL2 genes). The authors also performed an EMSA assay to show increased NFkb nuclear activity in the 3D model of psoriatic skin. Another finding is the increased level of IFNk.
The authors' conclusions imply that the model is valuable non only for target testing but also for target identification.
The manuscript is well written and organized and performed with well-described and sound methods. There are however a couple of major points that must be addressed:
1. T cell enrichment in this model is unclear to me. The is no physical separation between fibroblasts/keratinocyte layers and T cells. And these are heterologous T cells, that can produce a massive alloreactive responses, specially in several-days cutures. The level of HLA-matching between individuals can also influence the activation of T cells in the different samples.
I checked the previous work by the same authors and I did not find a clear explanation of this aspect.
Please address this point, because to my understanding this model mainly exposes keratinocytes to cytokines produced by activated T cells rather than including the T cell component. T cells should be autologous.
2. The figures do not show the control without T cells. This must be included.
3. The main findings: dysregulation of PTPRM and NELL2 as well as increase in IFNk have not been evidenced in the human pathology which has been extensively studied both at transcriptomic and single-cell level.
Why should the model reveal mechanisms that were not evidenced in the pathology?
Analysis of existing datasets on human psoriatic skin (ideally sc-RNAseq) should be included to support the importance of PTPRM and NELL2 in psoriasis.
4. ErK1/2 activation and in the role of NFkb have been described in psoriasis. What is the additional value of the results obtained in the 3D model (including EMSA)?
The model mainly emerges as a valuable preclinical tool to study pathogenic mechanisms or validate target molecules for therapeutic purposes.
Minor point:
-It is not necessary to mention common features between cancer cells and psoriatic keratinocytes (lines 496-497).
They conclude
Reviewer 3 Report
Geneviève Rioux et al. studied the cellular and molecular mechanisms involved in psoriasis pathogenesis using a useful T cells-enriched 3D psoriatic skin model which closely mimics native psoriatic skin. Two genes PTPRM and NELL2 deregulated in psoriatic substitutes were identified. The deregulation of these genes would support the activation of ERK1/2 pathway and increased phosphorylation of downstream NF-kB and Sp1. The manuscript is well prepared and experiments are scientifically conducted.
Please address below concerns before publication:
1. Any data has been done to compare gene expression patterns between this T-cell-enriched skin model and original patient skin samples? Will this new model provide more matches of gene expression than no T cell enriched one (54% match)?
2. Line 353. Do these genes differ significantly in any published RNAseq data of patient samples and normal skin samples as well? which can support the results in this 3D model.
3. Line 379, Any possible explanation for not detected of IFNk in day21?
4. Figures 1 and 2, please clearly clarify IL-17A neutralizing antibody or the RSK inhibitor were all added in PS+T? Did see the clarification for PS+RSK inhibitor in line 319, but not clear in line 317 for IL-17A. Suggest having a new label like PS+T+RSK inhibitor in figures.
5. Figure 4, any data support RSK inhibitor did work in this model, like decreasing the RSK expression in protein or RNA level?
Author Response
Please se the attachment.

Round 2
Reviewer 2 Report
The main concerns were not addressed
Author Response
We would like to thank the reviewer for the comments.
Reviewer 3 Report
The authors have addressed all the concerns.
Author Response

(The authors gave the same response as above.)
